# Autoencoder-based Detection of Insulin Pump Faults in Type 1 Diabetes Treatment

Elena Idi
Dept. of Information Engineering
University of Padova
Padova, Italy
idielena@dei.unipd.it

Francesco Prendin
Dept. of Information Engineering
University of Padova
Padova, Italy
prendinf@dei.unipd.it

Giovanni Sparacino
Dept. of Information Engineering
University of Padova
Padova, Italy
gianni@dei.unipd.it

Simone Del Favero
Dept. of Information Engineering
University of Padova
Padova, Italy
sdelfave@dei.unipd.it

*Abstract*—Individuals with type 1 diabetes (T1D) require life-long insulin replacement to compensate for deficient endogenous insulin secretion, which would otherwise result in abnormal blood glucose levels. In recent years, significant investments have been made to improve T1D management, leading to the widespread adoption of accurate technology such as continuous glucose monitoring (CGM) sensors and automated insulin delivery systems. However, malfunctions in these devices, particularly pump systems, can cause undesirable interruptions of insulin delivery posing significant safety risks if not promptly addressed. Due to the low frequency of these episodes, developing accurate algorithms to identify insulin pump faults remains a challenge. To address these issues, this paper proposes a novel approach for detecting insulin pump faults (IPFs) by combining the ability of a long short-term memory (LSTM) autoencoder to extract features, with the strength of random forest to distinguish between anomalous and normal patterns. This method was developed and evaluated using data from 100 subjects, simulated over 90 days with the UVa/Padova T1D Simulator, an FDA-approved nonlinear computer simulator of T1D physiology. In the test set, the proposed algorithm identified the 93% of the total faults, while raising 2 false alarms in 3 months on average. These findings suggest that deep learning algorithms can enhance the safety and reliability of insulin pump systems, contributing to more effective therapeutic technologies.

*Index Terms*—Fault Detection, Insulin Pump, Automatic Feature Extraction, Insulin Pump Occlusions.

## I. INTRODUCTION

Type 1 Diabetes (T1D) is a chronic metabolic disorder that affects millions of individuals worldwide [1]. Because of the autoimmune destruction of the pancreatic beta cells, this condition leads to a fundamental impairment in the body's ability to regulate blood glucose levels, requiring lifelong insulin replacement therapy. The daily management of T1D is complex and highly demanding for individuals, which have to frequently monitor their glucose levels, modify lifestyle behavior and compute tailored insulin injections. While excessive insulin administration can lead to hypoglycemia (i.e., glucose levels below 70 mg/dL), with the potential to trigger seizures, comas, and even death [2], on the other side, insufficient

insulin delivery could cause prolonged hyperglycemia (i.e., blood glucose levels exceeding 180 mg/dL). This condition is usually related to the development of chronic complications, including neuropathies, nephropathies, and cardiovascular diseases [3]. Furthermore, significant instances of hyperglycemia can also present an immediate risk to individuals, particularly when resulting from extended periods without insulin, leading to diabetic ketoacidosis (DKA), a life-threatening condition that requires patients' hospitalization.

In the last decades, technological advancements have significantly improved and revolutionized the management of T1D, introducing key innovative devices such as minimally invasive glucose sensors, known as Continuous Glucose Monitoring (CGM) sensors, that enable frequent blood glucose concentration measurements (at intervals of 1–5 minutes) without the need for finger pricks. Moreover, portable pumps, designed for continuous subcutaneous insulin infusion (CSII pumps), can facilitate the continuous insulin administration and allow the subjects to change insulin dosage throughout the day, also in real time [4]. Finally, closed-loop systems that adjust insulin infusion based on CGM readings have been developed and put on the market. These systems, usually referred to as artificial pancreas (AP), have shown clear potential to enhance the quality of glucose control while simultaneously reducing the actions requested to the patient to manage the disease [5].

Unfortunately, both CGM sensors and insulin pumps are susceptible to malfunctions, which can impact the overall functioning of the system and pose critical safety risks for the individual [6]. Insulin pump faults, such as infusion set blockages, mechanical failures and dislodgments, can result in undesirable insulin delivery interruptions to different degrees, including complete halts. If left untreated, these faults often lead to hyperglycemia and ketonemia [7], making pump issues one of the major contributors to diabetic ketoacidosis [8].

The early identification of IPF is a challenging task because insulin action on BG levels is not immediate and requires some time ($\sim$ 45 minutes). For this reason, the rises in glucose levels

related to the insulin absence (e.g. because of a pump occlusion), may become apparent on the glucose profile only some hours after the start of the malfunction. Nighttime occurrences of pump faults are particularly hazardous as individuals are often asleep and less prompt to intervention.

To deal with possible failures, some of the insulin pumps available in the market are indeed equipped with a self-monitoring system able to generate alerts to warn the patients when occlusions are suspected. Nevertheless, as reported in Gibney et al [9], there are still multiple undetected episodes, called *silent occlusions*: in particular the authors reported that respectively one-third of the patients that inserted the set with the applicator and one-half of the patients that inserted the set manually, experienced undetected insulin pump malfunctions.

*1) IPFs Detection, State of the Art:* Given the practical importance of the automatic identification of faults that affect insulin pumps, a number of studies in existing literature have addressed this issue. In particular, the problem was investigated by developing: heuristic methods [10], model-based approaches [11]–[15] and machine learning-based strategies [16], [17]. Howsmon et al. [10] developed a heuristic approach that defines three fault indicators by leveraging glucose and insulin data. Then, when these indicators simultaneously exceed previously tuned thresholds, an alert is raised to warn the subject that an IPF is on-going. Instead, in the model-based approaches presented in literature, the patient is intended as a dynamic system, where the output to control (i.e., glucose concentration) is affected by two inputs (insulin injection and meal intake). The model of the patient, that describes his physiology, is employed to predict future blood glucose levels, assuming that the insulin pump is properly working. Significantly large disparities between model predictions and CGM measurements indicate potential malfunctions. Different models can be used, including linear black-box models (as in [12]–[15] and physiological non-linear models (as in [11]). As limitation, the performance of the model-based approaches is strongly dependent on the quality of the model of which they leverage, and accurate models of T1D physiology are nontrivial to identify. To overcome this limitation, machine learning-based strategies [16], [17] have been developed. Both the approaches are based on the manual definition of features to detect anomalies in the system and are developed using in-silico data generated with the UVa/Padova T1D Simulator. Specifically, the main features employed are: i) the CGM signal, ii) the ratio between the trend of glucose concentrations and the carbohydrates on board (as in [18]), and iii) the ratio between insulin on board and carbohydrates on board. Concerning the machine learning models investigated, in [16], the authors employed multidimensional unsupervised anomaly detection techniques (i.e., Local Outlier Factor, Isolation Forest, and Histogram-based Outlier Scores). Of note, the proposed methods require a large amount of patient-specific data. Conversely, [17] examines both the previously mentioned unsupervised classifier and supervised classifiers. In accordance with the recent published guidelines [19], the approaches presented in [17] were developed in both population and personalized settings. Notably, while personalized strategies confirmed better performances, in case of limited patient-specific data the results suggested the adoption of population supervised classifier (with logistic regression and random forest as the two best options).

*2) Paper Contribution and organization:* In this work, we addressed the problem of the real-time IPF detection by developing a deep learning approach based on a recurrent autoencoder for the automatic feature extraction, and a random forest classifier aiming at distinguishing the anomalous state of the faults from the normal condition. While these methodologies are becoming popular tools for the identification of faults into the industrial and manufacturing field [20], their use within the T1D research community is at its early stage.

The paper is organized as follow: Section II describes the dataset used and reports the details of the simulation, while the methodology is described in Section IV. The results achieved, in accordance to the criteria presented in Section III, are presented and discussed in Section V and VI. Finally, some conclusions are provided in Section VII.

## II. DATASET

The pipeline proposed in this work is assessed using one of the latest versions of the UVA/Padova T1D Simulator [21], an FDA-accepted simulator of T1D physiology, developed for in silico testing prior to clinical trials.

Two synthetic datasets were generated, each consisting of 100 virtual subjects monitored over a period of 3 months. Everyday, three main meals were simulated, occurring at random times uniformly sampled within specific intervals: [06:30, 08:00] for breakfast, [11:30, 13:00] for lunch, and [18:00, 20:30] for dinner. The carbohydrate intake for each meal was uniformly sampled within the intervals [9-97g], [31-124g], and [28-140g], respectively. Basal insulin administration was regulated by a model predictive controller (MPC) [22], and insulin boluses were administered by the patient at each meal, based on estimated carbohydrate amounts.

This simulator version incorporates multiple sources of complexity such as the errors in carbohydrates estimation (as in [23]) and the intra-day insulin sensitivity fluctuations (as in [24]) for a more realistic scenario.

Blood glucose measurements, as well as insulin and meal data, were accessible at intervals of Ts = 5 minutes, a typical sampling frequency in Automated Insulin Delivery systems. The CGM sensor's measurement error was characterized according to [25]. Moreover, in one of the two datasets generated, one nocturnal fault per month is simulated, in accordance to the frequency reported in [26]. Each fault occurs on one randomly chosen day and is simulated by suspending insulin delivery for 6 hours, from midnight to 06:00. After six hours, it is assumed that patients become aware of the malfunction and resume insulin administration through manual intervention. These implementation choices align with established practices in the field and have been widely adopted in previous studies (e.g.; [10], [15], [16]). A summary of the populations' characteristics, detailing body

weight, age, percentage of time below range (i.e., below 70 mg/dL), in range (i.e., within [70-180] mg/dL) and above range (i.e., above 180 mg/dL) is reported in Table I.

TABLE I
DATASET SPECIFICS

| Metric | Data without IPF | Data with IPF |
|---|---|---|
| Body Weight [kg] | 75.2 (12.1) | 75.2 (12.1) |
| Age [years] | 33.8 (9.6) | 33.8 (9.6) |
| Time below range (TBR) [%] | 5.6 (5.2) | 5.6 (5.2) |
| Time in range (TIR) [%] | 76.1 (9.8) | 75.1 (9.7) |
| Time above range (TAR) [%] | 18.3 (9.6) | 19.2 (9.4) |

The data presented in this section, were divided as follow: the dataset without faults was used for the training of the autoencoder, while the dataset with faults was subjected to a training-test partitioning in 80:20 proportion for the application of the random forest (i.e., the algorithm was trained on 80 subjects and tested on the remaining 20). Since in real world scenarios training data might also be corrupted by anomalies, in a practical implementation, we envision to run preprocessing steps aiming to identify and discard all the data portion that can negatively affect the learning process of the autoencoder (AE). The summary of the data partitioning is reported in Table II.

TABLE II
SUMMARY: DATASET PARTITIONING

| Number of Subjects | Type of data | Step of the pipeline |
|---|---|---|
| Dataset1 100 subjects | No Faults | Autoencoder Training |
| Dataset2 80 subjects | 1 IPF/month | Random Forest Training and Threshold Selection |
| Dataset2 20 subjects | 1 IPF/month | Test of the Pipeline |

## III. EVALUATION CRITERIA

In accordance with [16], [17], the performance assessment relies on counting true positives (TPs), false negatives (FNs), and false positives (FPs). Specifically, if at least one alarm is triggered during an insulin pump fault, a TP is counted. Otherwise, a FN is recorded. Instead, a FP is considered if an alarm is erroneously raised in the absence of faults. Moreover, since a prolonged abnormal physiological state could persist for some hours after the restoration of insulin delivery, the alerts raised within a 6-hour window following insulin resumption (i.e., 6-12 hours from the start of the occlusion) are not counted as FPs.

Then, for each patient, we measured:

(i) the recall ($r$), also known as sensitivity: $r = \frac{TP}{TP+FN}$. representing the fraction of IPFs correctly detected.
(ii) the number of false positive per day ($fp/day$).
(iii) the detection time, computed as the time elapsed from the start of the IPF to the generation of the alarms.

Then, the average metrics over the dataset are considered and are denoted as $R$, $FP/day$ and $Delay$ in the following sections. It should be noted that, in this highly imbalanced dataset, where one fault arises per month (i.e. three faults for

each subject in the whole simulation), true negative occurrences and their related metrics (e.g., specificity) are of limited interest [27].

## IV. METHODS

### A. Fault detection Framework

The fault detection framework developed for IPFs detection combines the use of an autoencoder to automatically extract features from the data and a random forest for discriminating between anomalies and normal states. In particular, by learning the process to copy the inputs into the outputs, the autoencoder learns to compress and then reconstruct the input data, such that the hidden layers of the network, called *latent space*, can effectively capture the most relevant features needed for accurate reconstruction. In this way, the learned features can be seen as a representation of the underlying structure of the data and are employed as input to the random forest.

To this aim, we developed an ad-hoc detection pipeline, depicted in the lower panel of Figure 1, which is composed by 3 main steps:

1) *Data Preparation*: this step consists of the pre-processing, windowing and normalization of the inputs, that are then used as inputs of the autoencoder.
2) *Automatic Feature Extraction using the Autoencoder*: once fed by the input sequence, the latent space representation of the input is expected to provide meaningful information in presence of an IPFs, which represents an abnormal situation.
3) *Anomaly Detection and Alert Generation*: the features extracted from the previous step are used as input of a random forest classifier to perform the classification in anomalous or normal samples. When an anomaly is detected, an alert is raised to warn the patient that an insulin pump fault occurred.

### B. Step 1: Data Preparation

The available data, presented in Section II, includes readings from the CGM sensor and insulin injection amounts, obtained from the pump, as well as patient self-reported amount of CHO per meal. Since insulin boluses and meal intakes are impulse-like signals that can influence glucose levels for several hours, we introduce two new variables—Insulin on Board (IOB) and Carbohydrates on Board (COB). These variables, that have already been used in literature for several purposes (e.g. prediction [28], [29] and control [30]), carry information about the prolonged dynamics of insulin absorption and the gradual impact of carbohydrates on glucose levels, respectively.

In particular, the $IOB$ represents an estimate of the amount of insulin injected and not yet absorbed and is computed as the convolution of the injected insulin with a suitable exponentially decaying function provided in Schiavon et al [18]. Similarly, the $COB$ represents an estimate of the carbohydrates that the patient has eaten but are not absorbed yet and it is computed as in [18]. Moreover, to mirror the physiological absorption time required by insulin and carbohydrates to

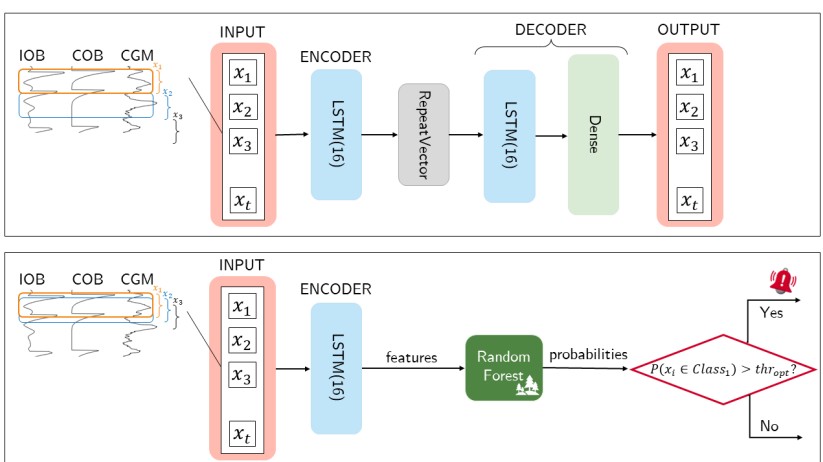

Fig. 1. In the upper panel is reported the autoencoder trained for the feature extraction. In the lower panel the pipeline for the detection is reported: in particular the encoder trained in used for the feature extraction and then Random Forest is applied for the classification and the alert generation.

impact on glucose levels, a delay is introduced in accordance with the literature [31] ($\sim$45 and $\sim$15 minutes respectively).

Then, in the context of multivariate time series reconstruction, we derived input and output sequences using a sliding window approach with a fixed window size $L = 1$ hour (12 samples). Finally, the data are normalized using a min-max scaler at population level using the training data of the autoencoder (according to Table II, Dataset 1).

### C. Step 2: Automatic Feature Extraction using the Autoencoder

In fields such as health informatics, autoencoders (AEs) have proven effective for automatically generating feature set without human intervention. For example, AEs can generate features that are difficult to extract in medical imaging [32] or can highlight complex patterns in the EEG signals [33]. The overall architecture is usually composed of two parts: an encoder and a decoder. The encoder compresses the input data into a lower-dimensional representation (the *latent space*), while the decoder reconstructs the original input from this compressed representation. Specifically, at a generic time $t$, the input of the encoder part is the sequence of CGM, IOB and COB defined as

$$X_t^{enc} = [x_{t-L+1}, x_{t-L+2}, ..., x_t] \in R^{3 \times L}$$

where $x_t = [CGM(t), IOB(t), COB(t)]$ and the output of the decoder is defined as

$$Y_t^{dec} = [\tilde{x}_{t-L+1}, \tilde{x}_{t-L+2}, ..., \tilde{x}_t] \in R^{3 \times L}$$

where $\tilde{x}_t$ is the reconstruction of $x_t$.

When dealing with time series data, Long Short-Term Memory (LSTM) neural networks can be of help in efficiently learning and maintaining dependencies (both on long and short-term) from sequences of input data. These networks belong to the category of recurrent neural network (RNN), but they overcome the vanishing-exploding gradient related issues affecting deep RNN during the training. The forget,

input, control and output gates are the key elements of the so-called memory cell of an LSTM and, at each time step, they control whether the incoming information is useful or if it must be erased from the cell. Therefore, an LSTM-based autoencoder represents a suitable approach for learning the complex dynamics characterizing the glucose-insulin system.

The architecture (summarized in the upper panel of Figure 1) has been implemented in Python (Keras library [34]) and consists of different layers, each playing a specific role in the learning process. The encoder component employs a LSTM layer with 16 units, capable to learn the temporal dependencies within the input data. Moreover, to enhance model generalization and prevent overfitting, a dropout layer with a dropout rate of 0.3 is added after the encoder. Subsequently, a repeat vector layer is employed to replicate the encoded features across time steps, facilitating the decoder's ability to generate sequential outputs. Finally, the decoder, i.e., a LSTM layer with 16 units, is used for input reconstruction, together with a dense layer to restore the dimensionality of the reconstructed output to match that of the original input. The Mean Absolute Error (MAE) is adopted as loss function to minimize the differences between the reconstructed outputs and the original data for the training.

Once the AE has been trained, the decoder part is removed, leaving only the encoder LSTM-based model. This encoder is then used to transform the input sequences (i.e., CGM, IOB and COB) into a set of features, which represent the input data in a reduced-dimensional space. Figures 2 and 3 (left panel) show the results at different steps of the detection pipeline: the output of the trained AE and the set of features extracted respectively.

### D. Anomaly Detection and Alert Generation

The encoder designed in the previous step can automatically extract 16 features from the input sequences, which are then used as inputs of a Random Forest (RF) classifier [35] to distinguish between normal or anomalous states. Specifically, the RF is trained on a subset of 80 subjects who experienced 3

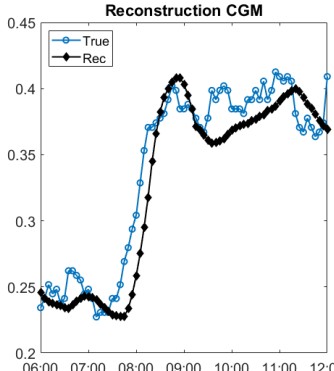 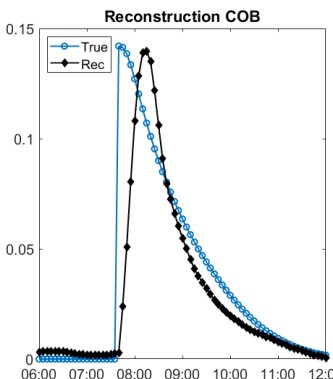 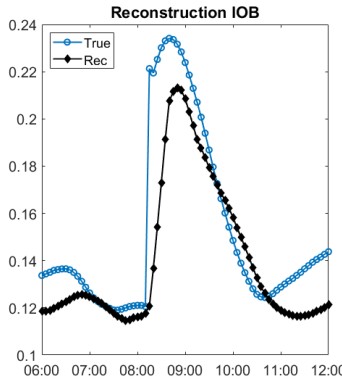

Fig. 2. Reconstruction of the inputs (CGM, COB and IOB respectively) on a training subject in absence of faults. The true signals are reported as blue solid line (circle markers), while the correspondent reconstructions are shown as linked black diamonds.

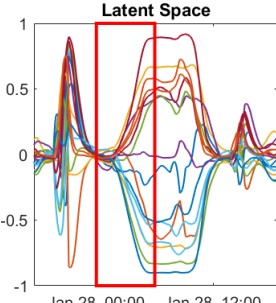 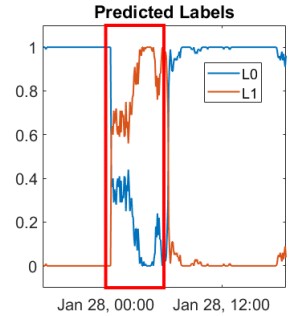

Fig. 3. In the left panel are reported the 16 features extracted with the encoder. The right panel shows the predicted probability computed by the Random Forest Classifier of belonging to that normal class ($L0$, in blue) or the anomalous one ($L1$, in orange). The red square indicates the occurrence of an insulin pump fault.

faults over 3 months of monitoring, with the aim to recognize the faulty samples among the data. In this framework, the output of RF is a probability score for each class, which represents the likelihood that a given instance belongs to the faulty or non-faulty class. By default, such a classification decision is based on the class with the highest probability score. However, selecting a suitable threshold is crucial for achieving accurate performance. To this end, the classification threshold ($thr_{opt}$) was fine-tuned following the procedure described in [16]. Briefly, a grid of possible thresholds is defined and a cost function $J$ is computed as

$$J(thr) = \sqrt{[1 - Recall(thr)]^2 + [FP/day(thr)]^2}.$$

$J$ quantifies the distance from the point where the ideal performance are achieved ($Recall = 1$; $FP/day = 0$).

For each threshold tested, the average recall and FP/day in the training set and the corresponding value of $J$ is computed and finally, the threshold that minimizes the cost function $J$ is selected as optimal: $thr_{opt} = argmin_{thr}(J(thr))$.
In this work, RF from the scikit-learn library [36] is used to compute the predicted class probabilities.

An example of the described probabilities is reported in the right panel of Figure 3 during an insulin fault.

## V. RESULTS

Figures 2 and 3 are generated to show the results at different steps of the detection framework. Particularly, Figure 2 reported the output of the AE where the reconstructed inputs are shown together with the original signals during 6 monitoring hours of a subject in the training set. The AE seems effective in recreating the inputs: indeed, deviations from the original signal are expected (e.g., CGM data are corrupted by measurement noise) but limited, and the AE is able to replicate the core dynamics. This holds true also for the COB and IOB signals which seem to accurately mimic the physiological absorption curves of carbs and insulin. Moreover, while in Figure 3 the left panel displays the set of features generated by the decoder, for a patient in the test set during an insulin pump fault, the right panel shows the probabilities calculated by the random forest of belonging to the normal class ($L0$) or the faulty one ($L1$). It can be seen that during a fault (highlighted with the red square), many of the extracted features spread towards the extreme values of 1 or -1. In accordance with that, the predicted probability of the data point to belong to the normal class decreases (from 1 to 0.4 and then to 0), while the probability of belonging to the anomalous class increases (from 0 to 0.6 and then to 1). A delay in the identification of the anomalous state is also visible in Figure 3 (right panel): in particular, the predicted probability of belonging to the anomalous class ($L1$) increases for some hours after the start of the pump occlusion. The delay depicted in the figure is reflected in the delay of the detection (reported in Table III), that represents the time required by the algorithm to recognize the fault. The detection results achieved on the test set after the computation of the threshold (on the training set) are summarized in Table III as mean and standard deviation. Moreover, the overall distributions of recall, FP/day and detection delay over the test set are reported in Figure 4: a scatter plot is also superimposed to the boxplot, so that each blue dot represents the performance in one test subject.

In the test set, the algorithm shows promising performance by achieving a recall of 0.93, meaning that the architecture is able to recognize almost the totality (93%) of the insulin pump

| | Recall [ ] | FP/day [ ] | Delay [min] |
|---|---|---|---|
| AE | 0.93 (0.17) | 0.02 (0.04) | 223 (67) |

faults, while raising on average 0.02 false alarms per day, equivalent to less than 2 false alarms in 3 months. Furthermore, the delay of detection is computed as the time elapsed from the start of the insulin suspension to the raise of an IPF alarm. Thus, the algorithm requires ∼220 minutes on average to recognize the IPF.

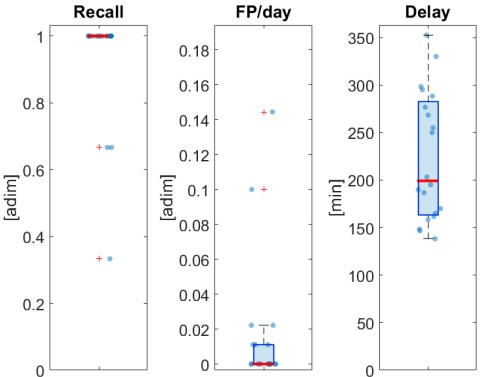

Fig. 4. Distribution of recall, FP/day and delay over the test set.

## VI. DISCUSSION

The promising obtained results in the test set by the proposed approach underscore the ability of the latent space representation to effectively extract critical features that are indicative of potential issues with the insulin pump. In the left panel of Figure 3, we presented the 16 features produced by the encoder during an episode of IPF. Most of these features exhibit a pronounced tendency to diverge rapidly towards extreme values when a fault episode occurs. This behavior suggests that the encoder is amplifying the anomalies in the data, particularly the deviations of input sequences from a normal situation. To better understand the role that these feature play on the following step (i.e.; RF classification), a feature importance analysis is conducted using the Mean Decrease in Impurity (MDI) method [37], which measures the importance of each feature based on how much it decreases the impurity in the RF model. In the context of RFs models, impurity measures how heterogeneous the data are at a given node. A high impurity means that the node contains a mix of data of different classes, while low impurity indicates that the data are more homogeneous, so that a node mostly contains data points of a single class. Since reducing impurity helps in making more accurate and reliable decisions in the model, Mean Decrease in Impurity method can provide insight on the importance of each feature in the classification process. Figure 5 shows the extracted features ordered according to their value of MDI importance.

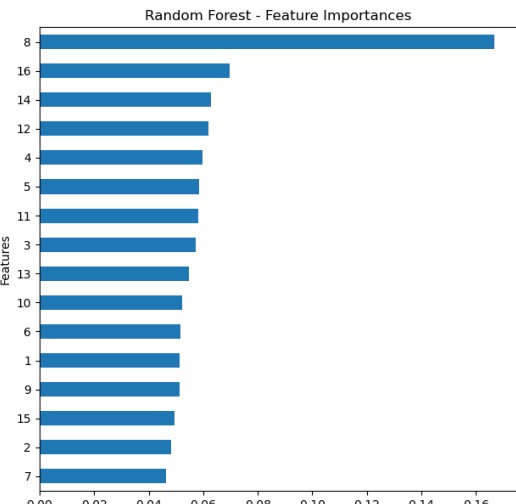

Fig. 5. Mean Decrease in Impurity (MDI) Importance. MDI importance value for each feature is shown in decreasing order. The larger the value, the more importance the model gives to that feature.

The top three identified features are also illustrated in Figure 6. This figure presents a 3D scatter plot of the features $[f_8(t), f_{16}(t), f_{14}(t)]$ for a subject in the test set, showing distinct clusters for faulty periods (red crosses) and normal conditions (dark circles). It is worth noting that the AE is also able to distinguish not only between faulty and non-faulty periods, but it can provide further insights about the restoration phase (blue diamond) which represent the 6 hours after an IPF.

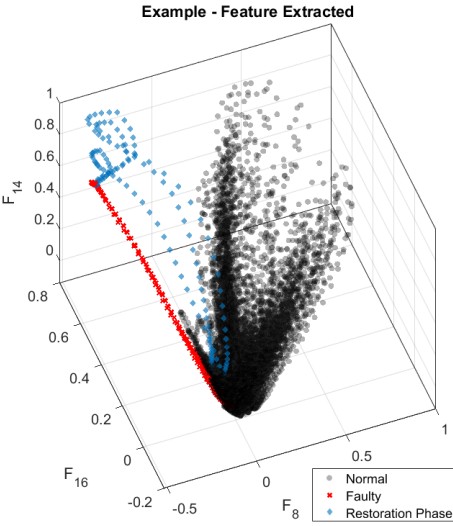

Fig. 6. 3D scatter plot of the features $[f_8(t), f_{16}(t), f_{14}(t)]$ of a subject belonging to the test set. The three features are reported as red crosses during an IPF, as dark circles in absence of faults and finally as blue diamonds during the restoration phase (the 6 hours after an IPF).

After the feature importance analysis, we investigate the performance achieved by the combination of the AE with the RF classifier in detecting the faults. The architecture achieves promising detection performances, as shown in Figure 4. In fact, the algorithm scores a recall equal to 1 for all the subjects,

with the exception of 3, and generates more than 0.1 false alarms per day for only one of the subjects in the test set.

Regarding the detection time, the recognition of the fault occurs in approximately 220 minutes on average. It's important to consider that, in the simulated scenario, the exact start of the fault is known and the impact of insulin absence becomes apparent on blood glucose only about 2 hours after the actual fault start. In view of this, the delay of detection obtained is in accordance with the study proposed in [38] by Klonoff et al where they induced real occlusions in clinical settings by applying a clip to the catheter of different insulin pumps available in the market and proved that the detection of the occlusion can take up to 4 hours.

To ensure a robust evaluation of the algorithm's performance across different subsets of the data, we employed a 5-fold cross-validation strategy. In this approach, the dataset is randomly partitioned into five equally sized folds and each fold is used as a test set once, while the remaining four folds are used for training. This process is repeated five times, with each fold acting as the test set exactly once. The results from each fold and the results averaged among the splits are presented in Table IV. Notably, the results across the different folds were consistent, indicating that the algorithm's performance is stable and not overly dependent on any specific subset of the data: our approach is able to recognize the 90% of the IPF on average while generating about 4 false alarms in 3 months.

TABLE IV
RESULTS OF THE K-FOLD CROSS VALIDATION

| Fold | Recall [ ] | FP/day [ ] |
|------|-----------|-----------|
| 1 | 0.93 (0.17) | 0.02 (0.04) |
| 2 | 0.81 (0.20) | 0.07 (0.10) |
| 3 | 0.98 (0.07) | 0.03 (0.06) |
| 4 | 0.85 (0.26) | 0.08 (0.10) |
| 5 | 0.95 (0.22) | 0.06 (0.09) |
| **Average** | **0.90** | **0.05** |

Finally, the obtained results are compared with the ones reported in the literature, summarized in Table V.

TABLE V
COMPARISON WITH THE STATE-OF-ART.

| Algorithm | Recall [ ] | FP/day [ ] | Dataset |
|-----------|-----------|-----------|---------|
| **AE-RF** | **0.93** | **0.02** | Simulator v2018 [21] |
| Random Forest [17] | 0.82 | 0.21 | Simulator v2018 [21] |
| IForest [16], [17] | 0.80 | 0.06 | Simulator v2018 [21] |
| Manzoni et al [15] | 0.91 | 0.12 | Simulator v2018 [21] |
| Herrero et al [39] | 0.80 | 0.08 | Simulator v2014 [40] |
| Howsmon et al [10] | 0.73-0.71 | 0.27-0.28 | Real data |

The proposed approach outperforms the method proposed in [17] where RF is applied on a set of manually extracted features. In particular, the new architecture can recognize more faults with a lower generation of false alarms. Similar results in terms of recall are achieved by [15] but with a higher amount of false alarms raised, while similar amount of false alarms are generated with the approaches proposed in [16], [39] that instead exhibit a lower recall. It should be take into account that, while [39] used an older version of the simulator (generating a scenario potentially less realistic and challenging), this work and [15]–[17] leveraged the same version of the simulator and the settings were consistent across the environments, thus allowing for a fair comparison. Finally, the heuristic method proposed in [10] appears to achieve worse results in terms of both recall and FP/day: however, this method is developed and assessed on real-world data.

## VII. CONCLUSIONS

Individuals with Type 1 Diabetes may potentially encounter a considerable safety risk if the insulin delivery is unexpectedly interrupted due to insulin pump faults or infusion set malfunctions. The automated identification of such issues can enhance the safety of T1D management systems, promoting trust toward therapeutic technologies.

In this work, we developed a novel approach that combines an LSTM-based autoencoder with a Random Forest classifier In particular, the AE was employed as an automatic feature extractor, that can capture the nonlinear relationships between the inputs (glucose concentrations, carbohydrates, and insulin administration). To do this, the AE was trained on faulty-free data to learn the essential features and then the encoder part was used for the automatic feature extraction in data containing faults. Finally, the features extracted from the latent space were used as inputs of a random forest classifier to distinguish patterns and anomalies in the data. The algorithm was developed and assessed using the UVa/Padova T1D Simulator, an FDA-accepted nonlinear computer simulator of T1D physiology. The obtained results are promising: almost the totality (0.93%) of the insulin pump faults is recognized, with less than 2 false alarms every 90 days. Despite employing one of the most realistic simulators available, simulated analysis are inevitably affected by simplifications. Therefore, the findings of this study should be validated through a larger dataset (possibly including other types of fault or different strategies for automatic control of insulin delivery), and in dedicated clinical trials, to test the robustness of the approach. Since the autoencoder is used as a black-box model, possible future studies could investigate the explainability of the latent space for a possible physiological interpretation of the features extracted. Furthermore, also the reconstruction errors of the inputs can introduce meaningful information for the detection of IPFs: even if the features extracted and the reconstruction errors are inevitably correlated, the quantification of the deviation of the reconstructed signals from the true ones can be of interest. Overall, the obtained results highlight the effectiveness of using deep learning techniques for IPF detection and underscore the potential of this approach to significantly improve the safety and reliability of T1D management systems.

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
