# OpenReview forum: "Autoencoder-based Detection of Insulin Pump Faults in Type 1 Diabetes Treatment"
_IEEE.org/EMBS/BHI/2024/Conference — IEEE BHI'24_

### Official Review · Reviewer_KPsp · 2024-07-26
**Autoencoder-based Detection of Insulin Pump Faults in Type 1 Diabetes Treatment**

**Overall Rating:** 8
**Confidence:** 5

**Other Quality Metrics:**

(a) Clarity of writing : great
(b) Clinical Significance : good
(c) Methodological Novelty : good
(d) Experiments and Results : good

**Questions For The Authors:**

- How was the min-max scaling performed ? usin the whole datasets, only the without faults one, or was it patient specific?
- Have similar strategies been considered for automatic control of insulin delivery?
- Are the results similar with several random 80:20 cuts?

**Strengths:**

The main contribution of this work is to transpose an autoencoder-based machine learning architecture used in industrial field to the insulin pump fault in T1D. The approach seems effective and better than other techniques used in the litterature.
The methods are well described and evaluation and results are adequately performed.

**Summary Of The Paper:**

In this paper, authors propose to detect insulin pump faults in T1 diabete using automatically extracted features in a autoencoder latent space and random forest classification. Two simulated dataset with/without faults were investigated.

**Weaknesses:**

The main limitation of this work, as mentioned by authors, is that working on simulated data does not enable one to project the performance of such a model on real data.
See Jacobs, P. G., Herrero, P., Facchinetti, A., Vehi, J., Kovatchev, B., Breton, M., ... & Mosquera-Lopez, C. (2023). Artificial intelligence and machine learning for improving glycemic control in diabetes: best practices, pitfalls and opportunities. IEEE reviews in biomedical engineering.

I propose here a list of suggestions that could improve the paper and readers' understanding.
- There is no need for subsection in the Introduction part.
- Competitive Machine learning approaches could be better described in the introduction section (what feature are used, which model, which data and what performance ?)
- Describe what patient speficifics were used for simulation
- Use more varied pumping fault scenarios. Why 6 hours only ?
- Correcting repetition about the description of autoencoder in IV.A and IV.C may help finding space
- Examples and illustrations provided in the method section can be moved in the results section:
"As an example of the ouput [...] and insulin"
"Figure 3 [...] fault"
- I would suggest to move MDI analysis and methods comparison into the results section and reducing conclusion section by integrating discussed topics into the discussion (i.e., real world, physiological interpretation). It is weird to see only one subsection in the discussion part.
- Just because I saw them, here are some typos/errors:
Figure 4: an empty legend on the ordinate
VI.A the results obtained --> the obtained results
a space is missing in VII "containing faults.Finally"

---

### Official Review · Reviewer_KKGy · 2024-07-28
**Autoencoder-based Detection of Insulin Pump Faults in Type 1 Diabetes Treatment**

**Overall Rating:** 6
**Confidence:** 3

**Other Quality Metrics:**

(a) Clarity of writing: excellent
(b) Clinical Significance: great
(c) Methodological Novelty: fair
(d) Experiments and Results: good

**Questions For The Authors:**

none

**Strengths:**

The paper is written in an easy-to-follow manner. The algorithm developed could be practically useful for detecting insulin pump failures.

**Summary Of The Paper:**

This work developed an abnormality detection approach for insulin pump fault detection. The authors first trained an auto-encoder to analyze three types of data relevant to insulin pumps: Continuous Glucose Monitoring, Insulin on Board, and Carbohydrates on Board. Features extracted by the encoder are then fed into a random forest-based classifier to identify abnormal statuses of the insulin pump.

**Weaknesses:**

While the combination of an LSTM-based autoencoder with a Random Forest classifier for insulin pump failure detection may represent a novel application, both components are fairly standard, resulting in low technical innovation. Additionally, although Table III summarizes results from state-of-the-art methods, these approaches were not applied to the same testing dataset, making the comparison less meaningful.

---

### Official Review · Reviewer_8MWz · 2024-07-31
**In this work, a novel approach for detecting insulin pump faults is proposed and evaluated using data from 100 subjects. A long short-term memory autoencoder was used, to extract features, with the strength of random forest, to distinguish between anomalous and normal patters.**

**Overall Rating:** 8
**Confidence:** 4

**Other Quality Metrics:**

(a) Clarity of writing – Excellent
(b) Clinical Significance - great
(c) Methodological Novelty - great
(d) Experiments and Results - great

**Questions For The Authors:**

Noting to point out.

**Strengths:**

Type 1 diabetes (T1D) is a chronic condition where the pancreas produces little or no insulin, necessitating the use of insulin pumps to regulate blood glucose levels. These devices play a critical role in maintaining the health and well-being of patients by delivering precise doses of insulin. However, faults in insulin pumps can lead to incorrect dosing, resulting in severe health complications. Detecting these faults promptly is crucial for patient safety.

**Summary Of The Paper:**

A novel approach that combines an LSTM-based autoencoder with a Random Forest classifier is proposed for detection of insulin pump faults in type 1 diabetes treatment.  The results are promising.

**Weaknesses:**

Although the results are promising, two false alarms happen every 90 days, and this is one result that should be in the future.

---

### Decision · Program_Chairs · 2024-09-23

Accept